# DISTRIBUTIONALLY ROBUST LEARNING FOR UNSUPERVISED DOMAIN ADAPTATION

## ABSTRACT

We propose a distributionally robust learning (DRL) method for unsupervised domain adaptation (UDA) that scales to modern computer-vision benchmarks. DRL can be naturally formulated as a competitive two-player game between a predictor and an adversary that is allowed to corrupt the labels, subject to certain constraints, and reduces to incorporating a density ratio between the source and target domains (under the standard log loss). This formulation motivates the use of two neural networks that are jointly trained — a discriminative network between the source and target domains for density-ratio estimation, in addition to the standard classification network. The use of a density ratio in DRL prevents the model from being overconfident on target inputs far away from the source domain. Thus, DRL provides conservative confidence estimation in the target domain, even when the target labels are not available. This conservatism motivates the use of DRL in self-training for sample selection, and we term the approach distributionally robust self-training (DRST). In our experiments, DRST generates more calibrated probabilities and achieves state-of-the-art self-training accuracy on benchmark datasets. We demonstrate that DRST captures shape features more effectively, and reduces the extent of distributional shift during self-training.

## 1 INTRODUCTION

In many real-world applications, the target domain for deployment of a machine-learning (ML) model can significantly differ from the source training domain. Furthermore, labels in the target domain are often more expensive to obtain compared to the source domain. An example is synthetic training where the source domain has complete supervision while the target domain of real images may not be labeled. Unsupervised domain adaptation (UDA) aims to maximize performance on the target domain, and it utilizes both the labeled source data and the unlabeled target data.

A popular framework for UDA involves obtaining proxy labels in the target domain through self-training (Zou et al., 2019). Self-training starts with a classifier trained on the labeled source data. It then iteratively obtains pseudo-labels in the target domain using predictions from the current ML model. However, this process is brittle, since wrong pseudo-labels in the target domain can lead to catastrophic failure in early iterations (Kumar et al., 2020). To avoid this, self training needs to be conservative and select only pseudo-labels with sufficiently high confidence level. This entails accurate knowledge of the confidence levels.

Accurate confidence estimation is a challenge for current deep learning models. Deep learning models tend to produce over-confident and misleading probabilities, even when predicting on the same distribution (Guo et al., 2017a; Gal & Ghahramani, 2016). Some attempts to remedy this issue include temperature scaling (Platt et al., 1999), Monte-Carlo sampling (Gal & Ghahramani, 2016) and Bayesian inference (Blundell et al., 2015; Riquelme et al., 2018). However, Snoek et al. (2019) has shown that the uncertainty estimation from these models cannot be trusted under domain shifts.

In this paper, we instead consider the distributionally robust learning (DRL) framework (Liu & Ziebart, 2014; 2017) which provides a principled approach for uncertainty quantification under domain shifts. DRL can be formulated as a two-player adversarial risk minimization game, as depicted in Figure 1(a). Recall that the standard framework of empirical risk minimization (ERM) directly learns a predictor $\hat{P}(Y|X)$ from training data. In contrast, DRL also includes an adversary $Q(Y|X)$

Figure 1: (a) Intuition of DRL under domain shift, where $X_s$ and $Y_s$ represent the source labeled data, and $X_t$ represents the target unlabeled data, $y_{\text{pred}}$ is the predictor's probabilistic labels and the $y_{\text{fake}}$ is the adversary's proposed probabilistic labels. (b) Architecture for end-to-end training of the DRL framework without class regularization. It is an instantiation of (a) using neural networks. The expected target loss cannot be evaluated due to lack of target labels in the UDA setting. Instead, we need to compute the gradients directly for training the networks. We present the details in Sec. 2.2.

that is allowed to perturb the labels, subject to certain feature-matching constraints to ensure data-compatibility. Formally, the minimax game for DRL is:

$$\min_{\hat{P}(Y|X)} \max_{Q(Y|X)} \text{loss}_{P_t(X)} \left( \hat{P}(Y|X), Q(Y|X) \right), \tag{1}$$

where the adversary $Q(Y|X)$ is constrained to match the evaluation of a set of features $\Phi(x, y)$ to that of the source distribution (see Section 2 for details). Note that the loss in (1) is evaluated under the target input distribution $P_t(X)$, and the predictor does not have direct access to the source data $\{X_s, Y_s\}$. Instead, the predictor optimizes the target loss by playing a game with an adversary constrained by source data.

A special case of UDA is the covariate shift setting, where the label-generating distribution $P(Y|X)$ is assumed to be the same in both source and target domains. Under this assumption, with log-loss and a linear predictor parameterized by $\theta$ and features $\Phi(x, y)$, (1) reduces to:

$$\hat{P}(y|x) \propto \exp \left( \frac{P_s(x)}{P_t(x)} \theta \cdot \Phi(x, y) \right). \tag{2}$$

Intuitively, the density ratio $P_s(x)/P_t(x)$ prevents the model from being overconfident on target inputs far away from the source domain. Thus, the DRL framework is a principled approach for conservative confidence estimation.

Previous works have shown that DRL is highly effective in safety-critical applications such as safe exploration in control systems (Liu et al., 2020) and safe trajectory planning (Nakka et al., 2020). However, these works only consider estimating the density ratio in low dimensions (e.g. control inputs) using standard kernel density estimator (KDE) and extending it to high-dimensional inputs such as images remains an open challenge. Moreover, it is not clear if the covariate-shift assumption holds for common high-dimensional settings such as images — which we investigate in this paper.

In this paper, we propose a novel deep-learning method based on the DRL framework for accurate uncertainties that scales to modern domain-adaptation tasks in computer vision.

**Summary of Contributions:**

1. We develop differentiable density-ratio estimation as part of the DRL framework to enable efficient end-to-end training. See Figure 1(b).
2. We employ DRL's confidence estimation in the self-training framework for domain adaptation and term it as distributionally robust self-training (DRST). See Figure 2.
3. We further combine it with automated synthetic to real generalization (ASG) framework of (Chen et al., 2020b) to improve generalization in the real target domain when the source domain consists of synthetic images.
4. We demonstrate that DRST generates more calibrated probabilities. DRST-ASG achieves competitive accuracy on the VisDA2017 dataset (Peng et al., 2017) with 1% improvement over the baseline class-regularized self-training (CRST) using the standard soft-max confidence measure.

5. We analyze the reason for the effectiveness of DRST through a careful ablation study.

One challenge for training DRL is that the training loss cannot be directly evaluated under the UDA framework. However, we show that the gradients of the target loss can indeed be evaluated. By deriving gradients for both neural networks and proposing a joint training algorithm (Alg. 1), we show the network can be trained efficiently.

We also directly incorporate class regularization in the minimax game under our DRL framework. This is a principled approach in contrast to standard label smoothing incorporated on top of a given learning method. In our ablation studies, we observe that the covariate-shift assumption progressively holds to a greater extent as the iterations in self-training proceed. This is also correlated with the greater ability to capture shape features through self training, as seen in the Grad-CAM visualization (Selvaraju et al., 2017).

## 2    PROPOSED FORMULATION AND ALGORITHMS

In this section, we first introduce the class regularized DRL framework (2.1) and then propose differentiable density ratio estimation to enable end-to-end learning of DRL using neural networks. We provide training details, especially the gradient computation for training both networks (2.2). This is unique for our setting since the actual training loss on target cannot be evaluated due to lack of target labels. Finally, we propose our self-training algorithm DRST in 2.3.

### 2.1    DISTRIBUTIONALLY ROBUST LEARNING WITH CLASS REGULARIZATION

We are interested in robustly minimizing the classification loss on the target domain with confidence of adversary's prediction regularized. We use a weighted logloss term to penalize high confidence in the adversary's label prediction as the regularization. We make the same covariate shift assumption as in (Liu & Ziebart, 2014) that only the marginal input distribution changes and $P(y|x)$ is shared between source and target: $P_s(x) \neq P_t(x)$ and $P_s(y|x) = P_t(y|x)$. We aim to solve the following:

$$\min_{\hat{P}(Y|X)} \max_{Q(Y|X)\in\Xi} \text{logloss}_{P_t(X)}\left(\hat{P}(Y|X), Q(Y|X)\right) - r\mathbb{E}_{P_t(x)Q(y|x)}[Y\log\hat{P}(Y|X)], \quad (3)$$

where $Y$ is a one-hot encoding of the class and $r \in [0,1]$ is a hyper-parameter that controls the level of regularization. In this formulation, the estimator player $\hat{P}(Y|X)$ first chooses a conditional label distribution to minimize the regularized logloss on the target domain and then an adversarial player $Q(Y|X)$ chooses a conditional label distribution from the set ($\Xi$) to maximize the regularized logloss. The constraint set $\Xi$ defines how much flexibility we want to give to the adversary. Usually, we design feature functions on both $X$ and $Y$ and restrict the adversary to match statistics of the expectation of the features. We have the following lemma:

**Lemma 1.** *If we choose feature map $\Phi(X,Y)$ as the statistics to constrain $Q(Y|X)$, then equation 3 can be reduced to a regularized maximum entropy problem with the estimator constrained:*

$$\max_{\hat{P}(Y|X)} \mathbb{E}_{P_t(x)\hat{P}(y|x)}[-\log\hat{P}(Y|X)] - r\mathbb{E}_{P_t(x)\hat{P}(y|x)}[Y\log\hat{P}(Y|X)] \quad (4)$$

$$\text{such that: } \hat{P}(Y|X)\in\Delta \text{ and } |\mathbb{E}_{P_s(x)\hat{P}(y|x)}[\Phi(X,Y)] - \mathbb{E}_{P_s(x,y)}[\Phi(X,Y)]| \leq \lambda,$$

*where $\Delta$ defines the conditional probability simplex that $\hat{P}(y|x)$ must reside within, $\Phi$ is a vector-valued feature function that is evaluated on input $x$, and $\mathbb{E}_{P_s(x,y)}[\Phi(X,Y)]$ is a vector of the expected feature values that corresponds with the feature function. $\lambda$ is the slack term of constraints.*

The proof of this lemma involves strong duality of the convex-concave function, such that the min and max player can switch the order. We refer more details to the appendix. The following theorem states the solution of the problem:

**Theorem 1.** *The parametric solution of (4) for $\hat{P}(y|x)$ takes the form:*

$$\hat{P}(y|x) \propto \exp\left(\frac{\frac{P_s(x)}{P_t(x)}\theta \cdot \Phi(x,y) + ry}{ry+1}\right), \quad (5)$$

*where the parameter $\theta$ can be optimized by maximizing the log-likelihood on the target distribution. The gradients take the form:*

$$\nabla_\theta \mathbb{E}_{P_t(x)P(y|x)}[-\log \hat{P}_\theta(Y|X)] = \mathbb{E}_{\tilde{P}_s(x)\hat{P}(y|x)}[\Phi(X,Y)] - \tilde{\mathbf{c}}, \qquad (6)$$

*where $\tilde{\mathbf{c}} \triangleq \mathbb{E}_{\tilde{P}_s(x)\tilde{P}(y|x)}[\Phi(X,Y)]$, as the empirical evaluation of the feature expectations.*

Here $\tilde{P}_s(x)$ and $\tilde{P}(y|x)$ are the empirical distribution. In principle, $P(y|x)$ is the ground truth conditional label distribution shared between source and target domains. We call $\mathbb{E}_{P_t(x)P(y|x)}[-\log \hat{P}_\theta(Y|X)]$ the expected target loss in the paper. Even though it is not available in practice, we can approximate the gradients (6) using the source data in training. The norm of the approximated gradient converges to the true gradient in the rate of $O(1/m)$, where $m$ is the amount of source data. The proof involves application of the Lagrangian multiplier, setting the derivative of each specific $\hat{P}(y|x)$ to 0 and utilizing the KKT condition. We refer details to the appendix.

We use this form to illustrate the property of representation-level conservativeness and the class-level regularization of our formulation.
**Representation-level conservativeness:** The prediction has higher certainty for inputs closer to the source domain, when magnitude of $P_s(x)/P_t(x)$ is large. On the contrary, if the inputs are farther away from the source, which means $P_s(x)/P_t(x)$ is small, the prediction is uncertain.
**Class-level regularization:** Hyper-parameter $r$ adjusts the smoothness of the $Q(Y|X)$'s label prediction in (3). It translates to the $ry$ terms in the parametric form. In training, we compute the gradients using source labels where $y$ is the one-hot encoding of the class. In testing, we can set $y$ to be all one vector to obtain smoothed confidence.

In machine learning methods using density ratios, such as transfer learning (Pan & Yang, 2009), or off-policy evaluation (Dudík et al., 2011), a plug-

---

**Algorithm 1** End-to-end Training for DRL

**Input**: Source data , Target data , DNN $\phi$),
DNN $\tau$ , SGD optimizer $Opt_1$ and $Opt_2$, learning rate $\gamma_1$ and $\gamma_2$, epoch number $T$.
**Initialization**: $\phi, \tau \leftarrow$ random initialization,
epoch $\leftarrow 0$
**While** epoch $< T$
 **For** each mini-batch
  Compute (10) back propagate
  Back propagate $\mathcal{L}_d$
  Optimizer $Opt_1(\gamma_1)$ updates $\beta$
  Compute $\hat{\mathbf{p}}$ following (5).
  Compute gradients using (9)
  Back-propagate through $\nabla_\phi \mathcal{L}_c$
  $\mathbf{w} = \mathbf{w} - \gamma_2 \cdot \nabla_{\mathbf{w}} \mathcal{L}_c$
  $\mathbf{b} = \mathbf{b} - \gamma_2 \cdot \nabla_{\mathbf{b}} \mathcal{L}_c$
  Optimizer $Opt_2(\gamma_2)$ updates $\alpha$
 epoch $\leftarrow$ epoch $+1$
**Output**: Trained $\alpha, \beta, \mathbf{w}, \mathbf{b}$

---

in estimator for the density ratio $P_s(x)/P_t(x)$ is used. However, density ratio estimation (Sugiyama et al., 2012), especially in the high-dimensional data, is rather different. It is also not the case that more accurate density ratio estimation would lead to the better downstream task performance. We have a synthetic example shown in Appendix E. To scale up the method for modern domain adaptation tasks, we ask the question: **can we train the density ratio estimation and the learning tasks that use the ratios together** such that they share the common goal–the target domain predictive performance?

## 2.2 DIFFERENTIABLE DENSITY RATIO ESTIMATION AND END-TO-END TRAINING FOR DRL

We propose an end-to-end training procedure for DRL such that density ratio estimation is trained together with the target classification. We use two neural networks for classification and the density ratio estimation, respectively. See figure 1(b).

**Differentiable density ratio estimation:** We make the observation that the density ratio in the parametric form (5) can be a trainable weight for each example, which can receive gradients from the objective function. On the other hand, the density ratio can be estimated using a binary classification using unlabeled source and target data. Therefore, we propose to train a discriminative neural network to differentiate the two domains for the density ratio estimation, which receive training signals from both the target classification loss and the binary classification loss.

**Expected target loss as a training objective:** According to Theorem 1, even though the expected target loss cannot be evaluated using data, we can approximate the gradient (6) using source samples. We can directly apply the gradients on the last layer of the classification network and back-propagate to the other layers. We next present the overall training objective and derive the gradients.

**The overall training objective:** Assume $\phi(x, \alpha, \mathbf{w}, \mathbf{b})$ is the classification neural net with parameters $\mathbf{w}$ and $\mathbf{b}$ in the linear last layer and parameters $\alpha$ in the other layers. The input is the source data. Assume $\tau(x, \beta)$ is the discriminative neural net with parameters $\beta$, with input as both unlabeled source and target data. The last layer of the classification network is: $\mathbf{w} \cdot \phi(x, \alpha) + \mathbf{b}$. We use the following training objective that accounts for the interactions between training density ratios and classification performance for the target domain:

$$\mathcal{L} = \mathcal{L}_c + \mathcal{L}_d = \mathbb{E}_{P_t(x)P(y|x)}[-\log \hat{P}(Y|X)] + \mathbb{E}_{\tilde{P}(x)\tilde{P}(c|x)}[-\log \hat{P}(C|X)], \qquad (7)$$

where $\mathcal{L}_c$ is the expected target loss $\mathbb{E}_{P_t(x)P(y|x)}[-\log \hat{P}(Y|X)]$, which cannot be evaluated but has gradients available for training from the DRL framework, $\hat{P}(Y|X)$ takes the following form with neural networks as feature functions: $\hat{P}(y|x) \propto \exp\left(\frac{\frac{P_s(x)}{P_t(x)}\mathbf{w}\cdot y\phi(x,\alpha)+\mathbf{b}+ry}{ry+1}\right)$. We use $\mathcal{L}_d$ to represent the cross-entropy loss on the domain discriminator that produces $\hat{P}(C|X)$, with $C \in \{\mathrm{s}, \mathrm{t}\}$ as the source domain class and target domain class. Here we use $\tilde{P}(x)$ and $\tilde{P}(c|x)$ to represent the overall unlabeled data distribution and the empirical domain class in the data. Then the predictions from the classifier are $\hat{P}(c|x) \propto \exp\{c\tau(x, \beta)\}$. Based on the Bayes' rule, the density ratio $\frac{P_t(x)}{P_s(x)}$ can be computed as:

$$\frac{P_t(x)}{P_s(x)} = \frac{P(x|\mathrm{t})}{P(x|\mathrm{s})} = \frac{P(\mathrm{t}|x)P(x)/P(\mathrm{t})}{P(\mathrm{s}|x)P(x)/P(\mathrm{s})} = \frac{P(\mathrm{t}|x)P(\mathrm{s})}{P(\mathrm{s}|x)P(\mathrm{t})}. \qquad (8)$$

Therefore, we can use a discriminator for estimating $\frac{P(\mathrm{t}|x)}{P(\mathrm{s}|x)}$ from unlabeled source and target data (Bickel et al., 2007).

**Gradients for the classification network:** With density ratios, given by the discriminative network, we compute the gradient of the classification network following (6) on $\mathbf{w}$, $\phi(x, \alpha)$, and $\mathbf{b}$.

$$\nabla_\phi \mathbb{E}_{P_t(x)P(y|x)}[-\log \hat{P}(Y|X)] = (\hat{\mathbf{P}} - \mathbf{y})\mathbf{w}; \qquad \nabla_\mathbf{w} \mathbb{E}_{P_t(x)P(y|x)}[-\log \hat{P}(Y|X)] = (\hat{\mathbf{P}} - \mathbf{y})\phi;$$
$$\nabla_\mathbf{b} \mathbb{E}_{P_t(x)P(y|x)}[-\log \hat{P}(Y|X)] = (\hat{\mathbf{P}} - \mathbf{y}); \qquad (9)$$

where $\hat{\mathbf{P}}$ and $\mathbf{y}$ are vectors of predicted conditional label probabilities using the current parameters as in Eq. (2.2) and the one-hot encoding of the true labels in the source data, respectively. Note that $\nabla_\mathbf{w} \mathbb{E}_{P_t(x)P(y|x)}[-\log \hat{P}(Y|X)]$ corresponds with the gradient for learning $\theta$ in Eq. (6), with $\Phi(x, y) = \mathbf{w} \cdot y\phi(x, \alpha) + \mathbf{b}$.

**Gradients of the discriminative network from expected target loss**: We denote $d_s = P(\mathrm{s}|x)$ and $d_t = P(\mathrm{t}|x)$ as the two weight scalars for each input $x$, and therefore we have $\hat{P}(y|x) \propto \exp\{\frac{d_s}{d_t}(\mathbf{w} \cdot y\phi(x, \alpha) + \mathbf{b})\}$. Since $\mathrm{vec}(d_s, d_t)$ is exactly the output of the density estimator, we treat $(d_s, d_t)$ as trainable variables and derive gradients from the expected target loss $\mathcal{L}_c$. In this way, the parameter $\beta$ of the density ratio estimation network has additional learning signals from both losses. We derive the gradients as:

$$\nabla_{d_s}\mathcal{L}_c = \frac{1}{d_t}\mathbb{E}_{\tilde{P}_t(x)\hat{P}(y|x)}[\mathbf{w} \cdot \hat{y}\phi(x, \alpha) + \mathbf{b}], \nabla_{d_t}\mathcal{L}_c = -\frac{d_s}{d_t^2}\mathbb{E}_{\tilde{P}_t(x)\hat{P}(y|x)}[\mathbf{w} \cdot \hat{y}\phi(x, \alpha) + \mathbf{b}], \quad (10)$$

which is dependent on unlabeled the target inputs $\tilde{P}_t(x)$, but does not rely on the target labels. We refer to the appendix for the detailed derivation. Then we concatenate (10) into a gradient vector and back-propagate the discriminative network. We summarize the procedure in Algorithm 1.

## 2.3 DISTRIBUTIONALLY ROBUST SELF-TRAINING

**Algorithm:** We propose Algorithm 2 to combine the DRL model with self-training. The idea is to regard each training epoch as a new distribution shift problem in DRL. After each training epoch, we make predictions on the target domain and select the top confident portion with the proxy labels to merge into the source training data. Both the pseudo labels and the model confidence are from the DRL model (5). Then the labeled source data and newly pseudo-labeled target portion become the new source set for the next learning epoch for DRL, as shown in Figure 2. As studied in the previous self-training work (Zou et al., 2018), the data proportion $p$ does not have significant impact on the results. After validating this in practice, we set $p = 0.5$ for DRST for simplicity.

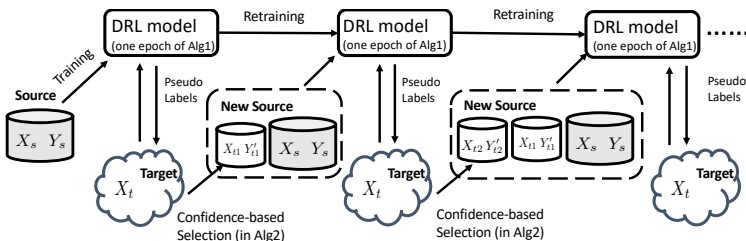

Figure 2: The self-training process of DRST, where $X_{t1}$ and $X_{t2}$ represent the portion of target data that is pseudo-labeled with $Y'_{t1}$ and $Y'_{t2}$. We use DRL model's prediction as pseudo-labels and its model confidence as the criterion for choosing pseudo-labeled data from the target domain.

**Initialization with ASG:** The quality of source model has an impact on the final self-training performance. Recently, (Chen et al., 2020b) proposed automated synthetic-to-real generalization (ASG) where ImageNet pre-trained knowledge is used to improve synthetic training through a distillation loss. It was shown that initializing with ASG trained model leads to significant improvement in self-training. As a result, we also take the ASG model as our model initialization, and show solid improvements from DRST on top of competitive baselines.

---

**Algorithm 2** Algorithm of DRST

**Input**: Source data , Target data, epoch number $T$, pseudo-label portion $p\%$.
Load pretrained model
**While** epoch $< T$
  Predict on target data using (5) and rank the confidence
  Label the top $p\%$ of target data
  Merge pseudo-labeled data into training data
  Conduct Algorithm 1 for one epoch
  epoch $\leftarrow$ epoch $+ 1$

---

# 3 EXPERIMENTS

In this section, we evaluate the performance of our method on benchmark domain adaptation benchmark datasets. We evaluate DRST as an effective unsupervised domain adaptation method (Sec. 3.1) and DRL as a domain generalization method providing more calibrated uncertainties (Sec. 3.2).

We adopt three datasets in our experiments: **Office31** (Saenko et al., 2010), **Office-Home** (Venkateswara et al., 2017) and **VisDA2017** (Peng et al., 2017). In particular, we use the largest data **VisDA2017** for the evaluation of DRST, for which we compare with (1) traditional UDA baselines: MMD (Long et al., 2015), MCD (Saito et al., 2018b) and ADR (Saito et al., 2018a); (2) recent self-training UDA baselines: CBST (Zou et al., 2020) and CRST (Zou et al., 2020); (3) other uncertainty quantification or UDA methods + self-training baselines: AVH (Chen et al., 2020a) + CBST and DeepCORAL(Sun & Saenko, 2016)+CBST. In addition, we use **Office31** and **Office-Home** for evaluating DRL's performance. We compare DRL with source training only and temperature scaling for demonstrating the calibration of the uncertainties used in DRST.

Apart from accuracy, we also use Brier score (Brier, 1950) and the realiability plot (Guo et al., 2017a) to evaluate the performance of our proposed method and the baselines. Brier score measures the mean squared difference between the predicted probability $p$ assigned to the possible outcome and actual outcome $y$: $BS = \frac{1}{n}\sum_{i=1}^{n}\sum_{j=1}^{m}(p_{ij} - y_{ij})^2$. The lower the Brier score is for a set of predictions, the better the predictions are calibrated.

## 3.1 UNSUPERVISED DOMAIN ADAPTATION USING DRST

**Accuracy and calibration:** Figure 3(a) and table 1 demonstrates that DRST outperforms all the baselines in accuracy. Our vanilla version of DRST outperforms CRST by more than 5% and with ASG initialization, we improve the SOTA self-training accuracy on VisDA by more than 1%. Note that in the rest of the results, we directly use DRST to represent DRST-ASG. Figure 3(b) demonstrates that DRST also achieves more calibrated confidence.

**Ablation study:** Figure 3(a)(b) also include two ablation methods. In the first ablation, we set $r$ to 0 so that there is no class regularization in DRL ("r = 0"). The prediction then follows the form in Eq. (2). In the second ablation, we set density ratio to 1 such that there is no representation level

Table 1: Accuracy comparison with different methods on VisDA2017.

| Method | Aero | Bike | Bus | Car | Horse | Knife | Motor | Person | Plant | Skateboard | Train | Truck | Mean |
|---|---|---|---|---|---|---|---|---|---|---|---|---|---|
| Source (Saito et al., 2018a) | 55.1 | 53.3 | 61.9 | 59.1 | 80.6 | 17.9 | 79.7 | 31.2 | 81.0 | 26.5 | 73.5 | 8.5 | 52.4 |
| MMD (Long et al., 2015) | 87.1 | 63.0 | 76.5 | 42.0 | 90.3 | 42.9 | 85.9 | 53.1 | 49.7 | 36.3 | 85.8 | 20.7 | 61.1 |
| MCD (Saito et al., 2018b) | 87.0 | 60.9 | **83.7** | 64.0 | 88.9 | 79.6 | 84.7 | 76.9 | 88.6 | 40.3 | 83.0 | 25.8 | 71.9 |
| ADR (Saito et al., 2018a) | 87.8 | 79.5 | **83.7** | 65.3 | 92.3 | 61.8 | 88.9 | 73.2 | 87.8 | 60.0 | 85.5 | 32.3 | 74.8 |
| CBST (Zou et al., 2020) | 87.2 | 78.8 | 56.5 | 55.4 | 85.1 | 79.2 | 83.8 | 77.7 | 82.8 | 88.8 | 69.0 | **72.0** | 76.4 |
| CRST (Zou et al., 2020) | 88.0 | 79.2 | 61.0 | 60.0 | 87.5 | 81.4 | 86.3 | 78.8 | 85.6 | 86.6 | 73.9 | 68.8 | 78.1 |
| AVH (Chen et al., 2020a) + CBST | 93.3 | 80.2 | 78.9 | 60.9 | 88.4 | 89.7 | 88.9 | 79.6 | 89.5 | 86.8 | 81.5 | 60.0 | 81.5 |
| DeepCORAL(Sun & Saenko, 2016)+CBST | 92.1 | 78.9 | 83.0 | 73.6 | 93.2 | 94.7 | 89.0 | 83.0 | 89.8 | 81.2 | 85.5 | 44.9 | 82.4 |
| **DRST** (proposed) | 93.47 | 86.30 | 65.74 | 68.03 | 93.99 | 95.08 | 87.34 | 83.30 | 92.97 | 88.65 | 83.66 | 66.42 | 83.75 |
| ASG (Chen et al., 2020b) | 88.81 | 68.55 | 65.31 | 78.06 | **95.78** | 9.11 | 84.89 | 29.58 | 82.13 | 33.76 | **86.00** | 12.04 | 61.17 |
| CBST-ASG (Chen et al., 2020b) | **95.12** | **86.53** | 79.83 | 76.01 | 94.61 | 92.34 | 85.94 | 75.08 | 89.23 | 82.16 | 73.42 | 56.49 | 82.23 |
| CRST-ASG (Chen et al., 2020b) | 92.38 | 81.30 | 74.63 | **84.40** | 90.90 | 92.43 | **91.65** | **83.78** | **94.92** | 88.12 | 74.88 | 61.10 | 84.21 |
| **DRST-ASG** (proposed) | 94.51 | 85.58 | 76.50 | 77.18 | 94.39 | **95.33** | 88.89 | 81.23 | 94.22 | **90.36** | 81.75 | 63.10 | **85.25** |

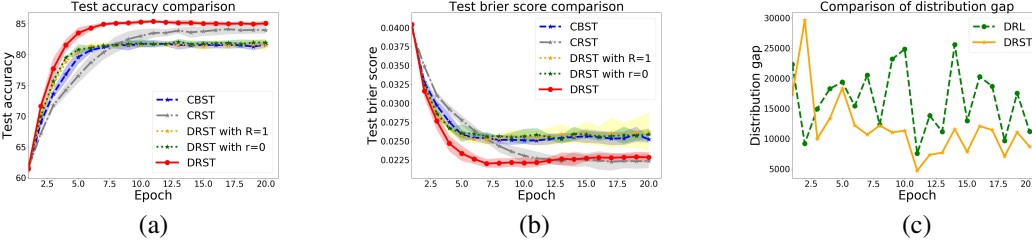

(a)         (b)         (c)

Figure 3: (a)- (b) Accuracy and Brier score of self-training on **VisDA2017** with 5 random seeds. We demonstrate the 95% standard error to show that DRST is outperforming the baselines significantly. (c) Distribution gap in $P(y|x)$: $P_s(x)/P_t(x) - P_s(x,y)/P_t(x,y)$, between source and target domains. Comparing with DRL without self-training, DRST helps reduce the gap.

consertiveness ("R = 1"), which means we mute the differentiable density ratio estimation in our method. We can see that DRST achieves the best performance when both components are present.

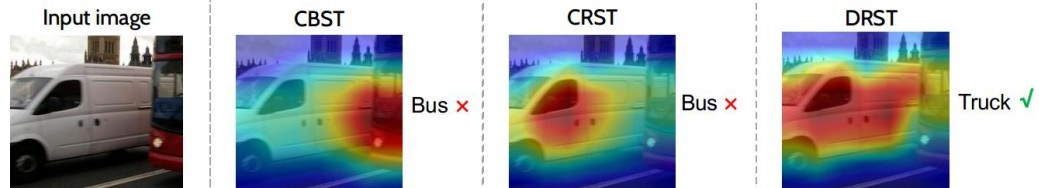

Figure 4: Model attention visualized using Grad-Cam (Selvaraju et al., 2017). We also show the predicted labels by different methods. Our method captures the shape features of the image better.

**Covariate shift:** In figure 3(c), we demonstrate the ratio difference between $P_s(x)/P_t(x) - P_s(x,y)/P_t(x,y)$, where $x$ is the last layer representations from each of the method. We estimate this ratio using discriminative density ratio estimators (per class). When $P_s(y|x) = P_t(y|x)$, the ratio difference is close to 0. We can see that the $P(y|x)$ gap between source and target is decreasing along self-training. This is due to the shape features captured by the method. We can also observe it from figure 4, where we visualize the last layer model attention of our model and the baselines. Therefore, even though the covariate shift assumption from DRL may not be satisfied at the beginning of self-training, self-training helps align $P(y|x)$ by promoting the learning of shape representations, such that $P_s(y|x)$ and $P_t(y|x)$ converges along the training.

**Density ratio estimation:** We show a pair of examples in the target domain with relatively high and low density ratios. In figure 5(a), a more typical "train" image obtains higher density ratio than a train on a busy platform, whose shape of the train is not obvious. DRL is able to give higher confidence to images that are better-represented in the synthetic training domain. More examples are given in the appendix.

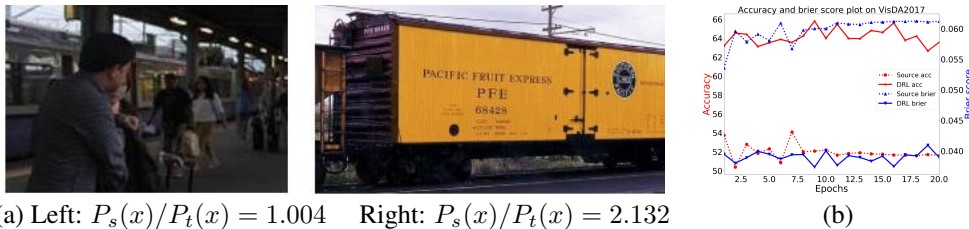

(a) Left: $P_s(x)/P_t(x) = 1.004$     Right: $P_s(x)/P_t(x) = 2.132$          (b)

Figure 5: (a) Density ratios and example target images for category "Train" in VisDA. Larger density ratio $(P_s(x)/P_t(x))$ indicates more certain prediction. DRL give more cerntain prediction on "train" that is better represented in the source domain, which on the right hand side. (b) Accuracy and brier score using DRL on VisDA. DRL significantly improves the source model along the training.

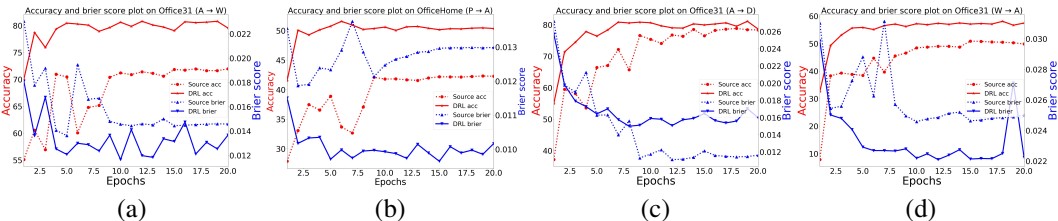

(a)                    (b)                    (c)                    (d)

Figure 6: Accuracy and Brier Score of DRL on **Office31** and **Office-Home**. DRL improves source-only by a large margin.

## 3.2    PREDICTIVE PERFORMANCE AND UNCERTAINTY QUANTIFICATION OF DRL

We compare DRL with source models on **VisDA2017**, **Office-31**, and **Office-Home** datasets to show the DRL model's accuracy and calibration. Note that DRL can be regarded as a lightweight model generalization method since the target unlabeled data is only used for the density ratio estimation.

**DRL's improvement on source training:** Figure 5 (b) and Figure 6 demonstrate the significant improvement over the source synthetic training model in all the data sets. Note that the only difference between the DRL and the source model is DRL's predictive form and the differentiable density ratio estimation. In these plots, all the models are trained with the same number of total training epochs. The accuracy and brier score at epoch 5 means we incorporate DRL to improve source training mode from epoch 5. Therefore, we can observe DRL's consistent significant improvement over the source training model even when starting late in the learning process. The lower brier score provides evidence on more conservative model confidence that benefits self-training.

**DRL's conservativeness in confidence:** Figure 7 demonstrates DRL's more calibrated uncertainty measure. We adopt the same definition of confidence and accuracy as (Guo et al., 2017a) and the same diagram plotting protocol as (Maddox et al., 2019). The closer the curve is to the dash line, the more calibrated the uncertainty is. DRL tends to be a little underconfident but also stays closer to the calibration line (dash line).

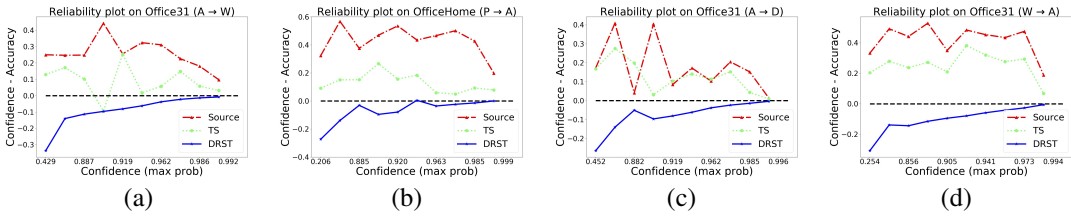

(a)                    (b)                    (c)                    (d)

Figure 7: Reliability diagrams with respect to source-only, temperature scaling (TS) and DRL. The data is separated into 20 bins with different interval lengths. The closer the curve is to the dash line, the more calibrated the uncertainty is. DRL is able to achieve more calibrated results, usually conservative confidence, while the other methods always tend to generate over-confident predictions.

## 4    RELATED WORK

**Uncertainty Quantification with Deep Models:** Bayesian methods largely contribute to current effort for complementing deep learning with uncertainty quantification. MCMC (Gal, 2016; Welling & Teh, 2011; Chen et al., 2014), (stochastic) variational inference for deep learning (Kingma & Welling, 2013; Louizos & Welling, 2017; Blundell et al., 2015; Riquelme et al., 2018), dropout variational inference (Gal & Ghahramani, 2015; Gal et al., 2017; Kendall & Gal, 2017), and SGD-based approximations (Mandt et al., 2017) aim to approximate the Bayesian inference from different perspectives in the deep learning setting. However, with a large number of parameters in the neural network, Bayesian deep learning may suffer from computational inefficiency. Other calibration methods such as temperature scaling (Platt et al., 1999; Guo et al., 2017b), deep ensemble (Lakshminarayanan et al., 2017), calibration regression (Kuleshov et al., 2018), quantile regression (Romano et al., 2019) and trainable calibration measure (Kumar et al., 2018). However, these methods are not applicable to our domain shift setting.

**Unsupervised Domain Adaptation:** One line of work in unsupervised domain adaptation methods focus on (feature) representation alignment between the source and target domains (Fernando et al., 2014; Pan et al., 2010; Sun et al., 2016; Wang et al., 2018; Baktashmotlagh et al., 2013; Gretton et al., 2012). In deep learning, it is more prevalent to use a discriminator network to differentiate two learned representations (adversarially) to locate such a feature space (Tzeng et al., 2017; Ajakan et al., 2014; Long et al., 2018; Sankaranarayanan et al., 2017). However, recent study show that this approach lacks theoretical grounding (Wu et al., 2019) and suffers from unstable training. Another line of work focus on more specific shift assumptions between two domains, like covariate shift (Shimodaira, 2000) and label shift (Lipton et al., 2018; Azizzadenesheli et al., 2019). Importance weighting on source samples is the common strategy to correct the shift. For the covariate shift case, even though various density ratio estimation has been investigated before (Sugiyama et al., 2012), only few works explore the possibility to apply them in high-dimensional data (Khan et al., 2019; Moustakides & Basioti, 2019; Park et al., 2020).

**Self-training for UDA:** Multiple self-training methods have been proposed to do unsupervised domain adaptation. Class-balance self-training (Zou et al., 2018) propose a class-balancing technique to avoid the gradual dominance of large classes on pseudo-label generation. Smoothing the model confidence further help improve the performance (Zou et al., 2019). It is also helpful to use a query selection strategy based on angular visual hardness (Chen et al., 2019). Recently, more theoretical understanding of self-training are developed (Kumar et al., 2020; Chen et al., 2020c). In this paper, we incorporate ideas from the CRST to smooth the DRL's model uncertainty and demonstrate that DRL's model confidence helps choose better pseudo-labeled data from the target domain.

**Distributionally robust learning:** Distributionally robust learning (Liu & Ziebart, 2014; 2017; Chen et al., 2016; Fathony et al., 2016; 2018) for domain shift is developed based on a pessimistic perspective of the finite samples existing in the source domain. However, the limitation of this line of work is that it usually only works for low-dimensional data. Another inconvenience lies in the requirement for the density ratio $P_s(x)/P_t(x)$ to be done beforehand. Instead of having the adversary as the conditional labeling function, an alternative is to have it perturbing the covariate variable (Ben-Tal et al., 2013; Shafieezadeh Abadeh et al., 2015; Sinha et al., 2017; Hu et al., 2018; Najafi et al., 2019). This line of work has been mostly focusing on model robustness against adversarial perturbations. Instead, in this paper, we focus on the benefit of DRL's uncertainty estimation for self-training in UDA. We apply neural networks and propose end-to-end learning method to handle high-dimensional image data.

## 5    CONCLUSION

In this paper, we develop a learning method under the distributionally robust learning framework for modern domain adaptation. We propose differentiable density ratio estimation and class regularization in the framework. We develop end-to-end training techniques for our proposed method. Using DRL's model confidence, our self-training algorithm achieves SOTA predictive performance while stay calibrated. We also demonstrate that self-training helps reduce the distribution gap between source and target domains, facilitating DRL to be effective.

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

## A    PROOF OF LEMMA 1

*Proof.* The two-player game in (3) can be written as:

$$\min_{\hat{P}(Y|X)} \max_{Q(Y|X)\cap\Xi} -\sum_x P_{\mathrm{t}}(x) \sum_y Q(y|x) \log \hat{P}(y|x) - r \sum_x P_{\mathrm{t}}(x) \sum_y Q(y|x) \mathbf{I}(\arg\max_{y'} Q(y'|x) = y) \log \hat{P}(y|x).$$

According to strong Lagrangian duality, we can switch the order of the two players and it is equivalent to:

$$\max_{Q(Y|X)\cap\Xi} \min_{\hat{P}(Y|X)} -\sum_x P_{\mathrm{t}}(x) \sum_y Q(y|x) \log \hat{P}(y|x) - r \sum_x P_{\mathrm{t}}(x) \sum_y Q(y|x) \mathbf{I}(\arg\max_{y'} Q(y'|x) = y) \log \hat{P}(y|x).$$

Solving the minimizing problem first assuming that we know $Q(Y|X)$, we get the result that $\hat{P}(Y|X) = Q(Y|X)$. Plugging it into the maximizing problem, the whole problem reduces to (4). $\qquad\square$

## B    PROOF OF THEOREM 1

*Proof.* The generalized constrained optimization problem in (4) can be written as:

$$\max_{\hat{P}(Y|X)} -\sum_{x\in\mathcal{X}} P_{\mathrm{t}}(x) \sum_{y\in\mathcal{Y}} \hat{P}(y|x)(\mathbf{1} + r\mathbf{I}(\arg\max_{y'} \hat{P}(y'|x) = y)) \log \hat{P}(y|x)$$

$$\text{such that: } \sum_{x\in\mathcal{X}} \sum_{y\in\mathcal{Y}} P_{\mathrm{s}}^k(x) \hat{P}(y|x)[f_k(X,Y)] = \tilde{c}_k', \forall k \in \{1,...,K\}$$

$$\forall x \in \mathcal{X}: \sum_{y\in\mathcal{Y}} \hat{P}(y|x) = 1$$

$$\forall x \in \mathcal{X}, y \in \mathcal{Y}: \hat{P}(y|x) \geq 0.$$

Note that the final constraint is superfluous since the domain of the objective function is the positive real numbers. The Lagrangian associated with this problem is:

$$\mathcal{L}(\theta,\lambda) = -\sum_{x\in\mathcal{X}} P_{\mathrm{t}}(x) \sum_{y\in\mathcal{Y}} \hat{P}(y|x)(\mathbf{1} + r\mathbf{I}(\arg\max_{y'} \hat{P}(y'|x) = y)) \log \hat{P}(y|x) + \tag{11}$$

$$\sum_k \theta_k \left[ \sum_{x\in\mathcal{X}} \sum_{y\in\mathcal{Y}} P_{\mathrm{s}}^k(x) \hat{P}(y|x) f_k(x,y) - \tilde{c}_k' \right] + \sum_{x\in\mathcal{X}} \lambda(x) \left[ \sum_{y\in\mathcal{Y}} \hat{P}(y|x) - 1 \right],$$

where $\theta$ and $\lambda(x)$ are the Lagrangian multipliers. Taking the derivative with respect to $\hat{P}(y|x)$,

$$\frac{\partial}{\partial \hat{P}(y|x)} \mathcal{L}(\theta,\lambda) = -P_{\mathrm{t}}(x)(\mathbf{1} + r\mathbf{I}(\arg\max_{y'} \hat{P}(y'|x) = y)) \log \hat{P}(y|x) - P_{\mathrm{t}}(x) + \sum_k P_{\mathrm{s}}^k(x) \theta_k f_k(x,y) + \lambda(x),$$

setting equal to zero, $\frac{\partial}{\partial \hat{P}(y|x)} \mathcal{L}(\theta,\lambda) = 0$, and solving, we get:

$$(\mathbf{1} + r\mathbf{I}(\arg\max_{y'} \hat{P}(y'|x) = y)) \log \hat{P}(y|x) = -1 + \sum_k \frac{P_{\mathrm{s}}^k(x)}{P_{\mathrm{t}}(x)} \theta_k f_k(x,y) + \frac{\lambda(x)}{P_{\mathrm{t}}(x)} + \frac{(\mathbf{1} + r\mathbf{I}(\arg\max_{y'} \hat{P}(y'|x) = y))}{P_{\mathrm{t}}(x)}.$$

Therefore, if we instead set $y$ as a one-hot encoding of the prediction, we conclude:

$$\hat{P}(y|x) \propto e^{\frac{\sum_k \frac{P_{\mathrm{s}}^k(x)}{P_{\mathrm{t}}(x)} \theta_k f_k(x,y) + ry}{1 + ry}},$$

The derivation of gradient then resembles the derivation in Theorem 1 in (Liu & Ziebart, 2014).

$\qquad\square$

Source:

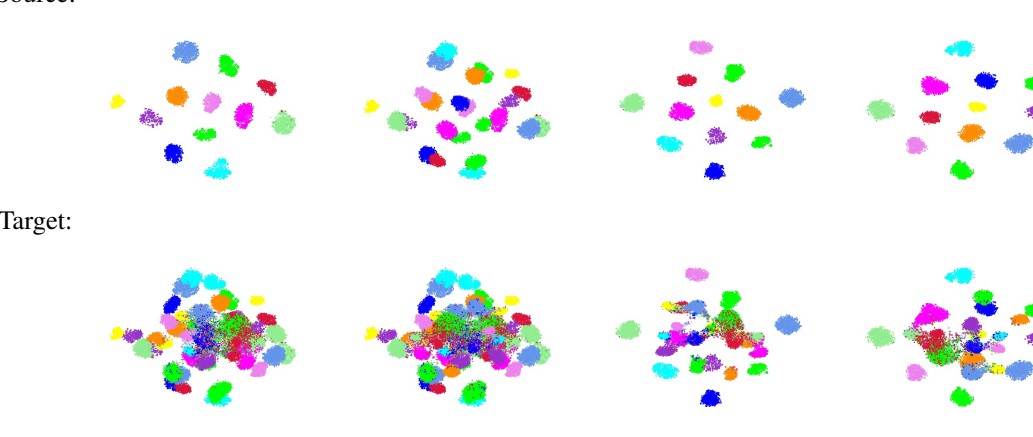

Target:

ASG          ASG+CBST         ASG+CRST         ASG+DRST

Figure 8: TSNE visualization of the learned classifier using different methods. Using DRST, the classes are well-separated.

## C  DETAILED DERIVATION OF GRADIENTS

Gradient of the expected target loss over source densities $d_s$, where we use $R$ for the density ratio $\frac{P_s(x)}{P_t(x)}$ or $\frac{d_s}{d_t}$:

$$
\begin{aligned}
\frac{\partial L(\theta)}{\partial d_s} &= \frac{\partial}{\partial d_s}(-\theta \mathbb{E}_{P_s(x)P(y|x)}[\Phi(X,Y)] + \mathbb{E}_{P_t(x)}[\log Z_\theta(X)]) = \mathbb{E}_{P_t(x)\hat{P}(y|x)}[\theta\Phi(X,Y)/d_t] \\
&= \frac{1}{d_t}\frac{\partial L(\theta)}{\partial R},
\end{aligned}
\tag{12}
$$

and gradient of loss over target densities $d_t$:

$$
\frac{\partial L(\theta)}{\partial d_t} = \frac{\partial}{\partial d_t}\mathbb{E}_{P_t(x)}[\log(Z_\theta(X))] = -\frac{d_s}{d_t^2}\mathbb{E}_{P_t(x)\hat{P}(y|x)}[\theta\Phi(X,Y)] = -\frac{d_s}{d_t^2}\frac{\partial L(\theta)}{\partial R}, \tag{13}
$$

where:

$$
\frac{\partial L(\theta)}{\partial R} = \mathbb{E}_{P_t(x)}\left[\sum_{y\in Y}\frac{\exp(R,\theta,\Phi)}{Z_\theta(X)}\theta\Phi(X,Y)\right] = \mathbb{E}_{P_t(x)\hat{P}(y|x)}[\theta\Phi(X,Y)]. \tag{14}
$$

## D  ADDITIONAL DRST EXPERIMENTAL RESULTS

We demonstrate the TSNE plot of the learned decision boundaries for CBST, CRST and DRST in figure 8. Figure 9 shows the misclassification entropy comparison between source model and DRL model. Misclassification entropy is calculated as $S_i = \frac{1}{n}\sum_{i=1}^{n}\sum_{j=1}^{m}p_{ij}\log p_{ij}$ , where $n$ is the number of samples and $m$ is the number of categories in the dataset, and $p_{ij}$ indicates the prediction probability of the $i$th sample on the $j$th category. The larger misclassification entropy is, the more uncertain the model prediction result is on the wrong predictions. This means the model would fail more gently. Figure 10 demonstrates additional model attention visualization. Figure 11 shows additional target examples with high and low density ratios. Our model is able to find noisy and less-represented image from the target data by estimating the density ratios. DRL would be more uncertain on those data.

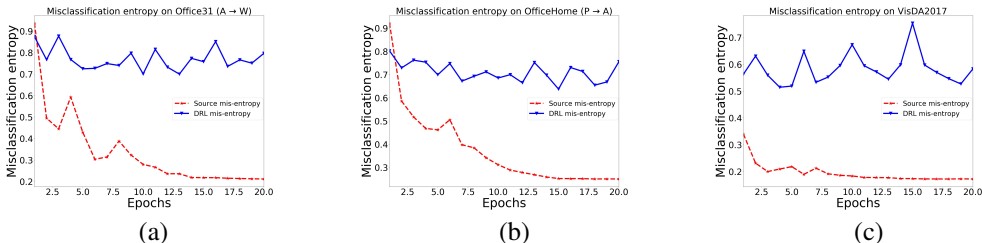

Figure 9: Comparison of DRL and DRST of misclassification entropy on different datasets.

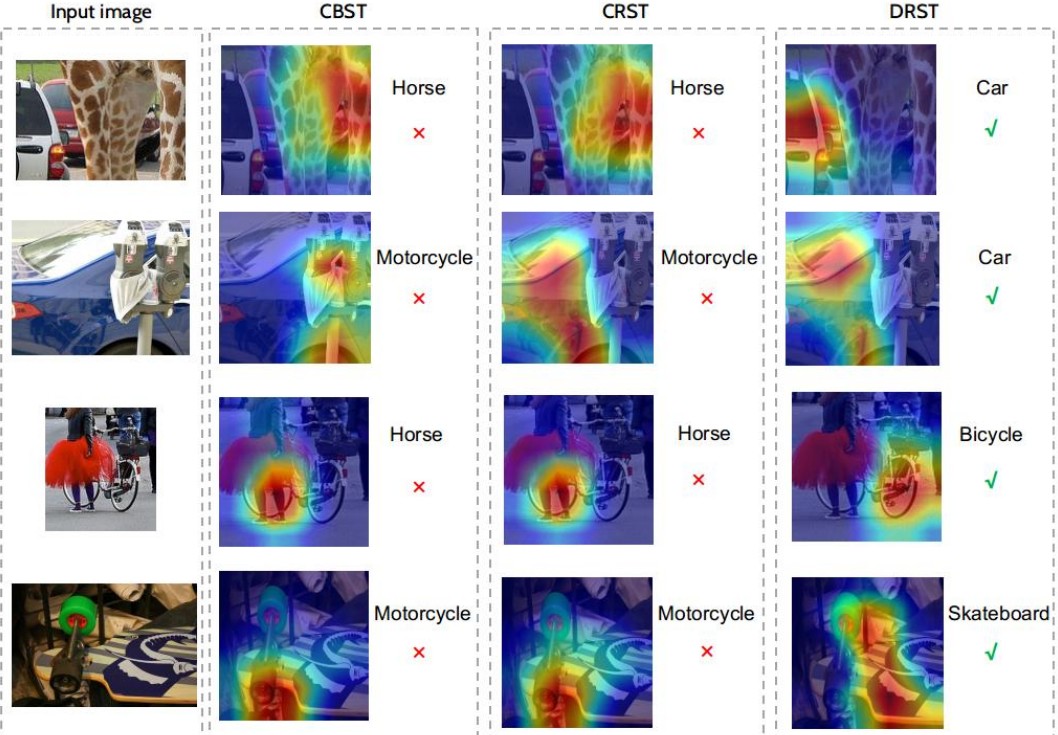

Figure 10: Model attention visualized using Grad-Cam (Selvaraju et al., 2017). Our method can also capture the domain knowledge well. For example, the first row input image contains a giraffe and a car. However, giraffe is not a existing label for VisDA2017. While CBST and CRST captures the wrong information, DRST is able to correctly capture the domain information.

# E    SIMULATION ON PLUG-IN ESTIMATOR

DRL for domain shift requires $P_s(x)/P_t(x)$ to adjust the representation-level conservativeness. Like many other machine learning methods using importance weights, like in transfer learning (Pan & Yang, 2009), or off-policy evaluation (Dudík et al., 2011), we can use a plug-in estimator for the density ratio $P_s(x)/P_t(x)$. However, density ratio estimation (Sugiyama et al., 2012), especially in the high-dimensional data, is rather different. Here, we ask the question: **whether a more accurate density ratio estimation leads to greater predictive performance in the downstream tasks?**

We show a two-dimensional binary classification example in figure 12 to demonstrate relation between the performance of the density (ratio) estimation and the performance of the ultimate target learning tasks. We use the RBA method (Liu & Ziebart, 2014) as an example. We conduct kernel density estimation (KDE) and evaluate the average log-likelihood of the source and target domain. We take the ratio of the density from KDE and plug in the RBA method. We can see that the case with higher log-likelihood actually fail to give informative predictions. One of the reason is the

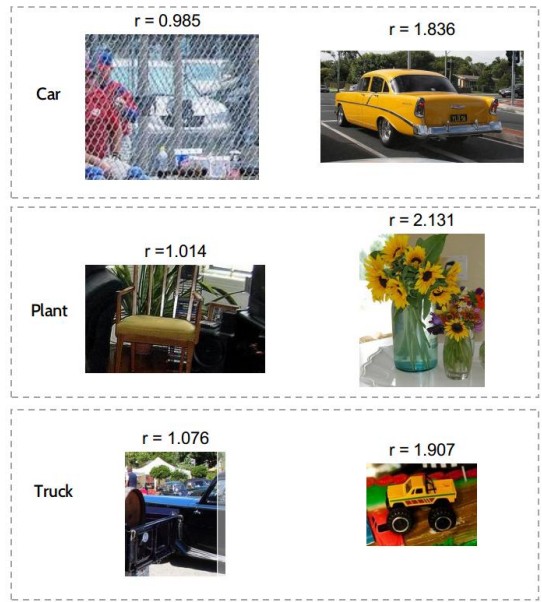

Figure 11: Additional examples for density ratio estimation for different categories. We can observe that data less well-represented in the source data has much lower density ratio. This shows that our learned density ratio is a good measure of the level of representation of data in source and target domains.

density (ratio) estimation task, as an independent learning task, does not share information with the downstream prediction tasks that use the ratios.

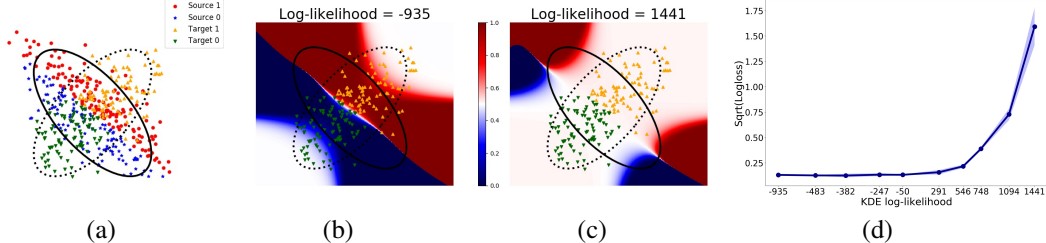

Figure 12: (a) Source and target data points are drawn from two Gaussian distributions. Solid line: source and dashed line: target. The underlying true decision boundary for the binary classes is the same between the two domains. (b)-(c) Prediction with density ratios from low and high density estimation likelihoods. With more accurate density estimation, the RBA predictor gives overly conservative predictions on the target domain. The colormap is the confidence $P(\text{"1"}|x)$. (d) With larger likelihood in density ratio estimation, the target log loss becomes worse.

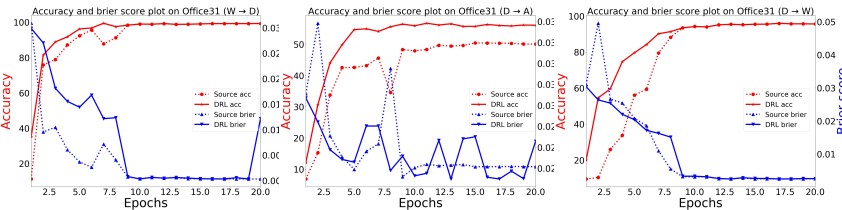

Figure 13: Additional Office31 results on DRL compared with source-only, which is a complement for Figure 6.

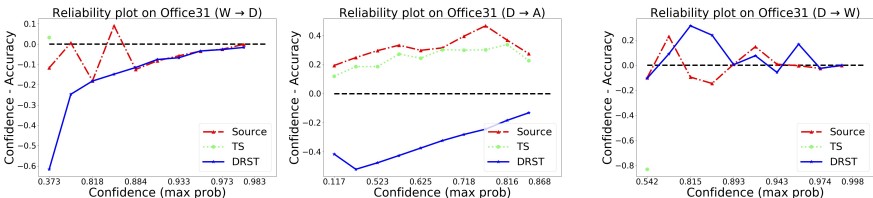

Figure 14: Additional Office31 results on reliability plots, which is a complement for Figure 7. DRL is compared with source-only and temperature scaling.

## F    ADDITIONAL RESULTS FOR DRL ON OFFICE-31

We include additional result for distributionally robust learning on office-31 in Figure 13 and Figure 14.

## G    CODE REPOSITORY

Code    can    be    found    at:    `https://anonymous.4open.science/r/`
`2ed8a9ce-f404-4489-9ef7-a3a83e02a44c/`

