# OpenReview forum: "Distributionally Robust Learning for Unsupervised Domain Adaptation"
_ICLR.cc/2021/Conference — Reject_

### Official Review · AnonReviewer4 · 2020-10-26
**Well motivated idea and interesting formulation, but lacking clarity around the main components and incomplete evaluation.**

**Rating:** 6
**Confidence:** 3

**Review:**

I find the paper to be well motivated. Self-labeling has proven to be a useful approach for unsupervised domain adaptation. And since wrong pseudo-labels in the target domain result catastrophic failure in early iterations, it makes sense to calibrate the production of pseudo-labels through the use of uncertainty estimation. This is done through the framework of distributionally robust learning.

I find three central concepts in the method: First, the formulation of the problem as a min-max game between the adversary and the predictor (appears in Eq.1 and Eq.3). Second, the relaxation of the predictor into a formula involving a ratio of densities (Eq.2, Eq.4 and Eq.5) and third the estimation of the gradients of the target loss through the use of source data. While the individual components seem well motivated and clear, I struggle to see the connection between them. For instance, how does figure 1(a) related to figure 1(b)? What is the intuition of using a ratio of densities as a good uncertainty estimation? How can source data be used to approximate gradients on the target? A clearer explanation of the connection could be useful or at least a sketch of the proofs be provided in the main text.

In addition, while Alg.1 is shown to be a relaxation of the original formulation, Alg.2 seems to be an adhoc attempt to combine the DRL framework with self-training. Why is this the correct formulation to use? At the very least, an experimental comparison to other possible approaches can be used - for instance, one can apply a standard UDA approach instead of Alg.1 and choose, based on standard confidence measures, the top confident labels for a subsequent iterations of Alg.2.

The use of three different datasets though a variety of experiments demonstrates the usefulness of the approach.  However, some experimental evaluation is missing in my view. First, an ablation study for the different components of both Alg.1. and Alg.2 would be usefull. For instance, in Alg.1, viewing the density ratios as an uncertainty estimation, what is the effect of using other approximations for the uncertainty, either by the adjusting this ratio, or using existing uncertainty estimation methods. How does a different choice for the flexibility of the adversary affect UDA results (Eq.3)  For Alg.2 what is the effect of using different pseudo-label portion p?

Further, the paper claims to better captures the shape features, but this is not shown.  One could use the evaluation shown in [1], for instance, to demonstrate this.

Overall, on the positive side,  I found the problem well motivated and important. The individual components of Alg. 1 are well explained and shown theoretically. The overall approach well evaluated on 3 datasets and some of the components evaluated.
On the negative side, clarity of the paper could be improved, particularly with regards the the connection between the main components. Additional experiments, and an in particular an ablation study regarding Alg1. and Alg.2 would provide a much better understanding of the method.

[1] ImageNet-trained CNNs are biased towards texture; increasing shape bias improves accuracy and robustness. ICLR 2019.

---

> ### Author Response · Authors · 2020-11-23
> **Author response to AnonReviewer4**
>
> Thanks a lot for your summary of our contribution and the central concepts in our paper! In addition to the general response, we address your concerns as follows:
>
> **Regarding the formulation:**
>
> The connection between the first and second concepts (mini-max formulation and the parametric formulation associating a density ratio) is that “the second concept” is the solution of the minimax formulation in “the first concept” when the feature constraint set for the adversary is an affine function of Q(y|x) and the loss function is a convex-concave function. We had the proof of Lemma 1 and Thm 1 in appendix A and B, due to the space limit. We added a brief description of the proof back to the paper now. Then the idea of training the density ratio in the same formulation is due to the suboptimality of a pretrained (plug-in) density ratio. We illustrate it in appendix E. We used a toy example to show that a  more accurate density ratio estimation may not lead to a greater predictive performance in the downstream tasks. This motivates us to train both components together. We added a short paragraph back to the end of Section 2.1 now.
>
> Then to answer your specific questions:
>
> * “How does figure 1(a) and (b) connect with each other?” Figure 1(a) is the general minimax game formulation for DRL. Figure 1(b) is a specific form of implementation of the method when we instantiate the classifier learning and density ratio estimation using deep neural networks. So in short, Figure 1(b) is the solution for a specific version of Figure 1(a) when the “feature constraints” in (a) are incorporating neural networks. We edited the caption of Figure 1 to make this connection more explicit.
>
> * “Why use a density ratio for uncertainty estimation?” More precisely, our density ratio is adjusting the uncertainty in addition to the features and parameters in traditional softmax formulation. Our formulation is the solution of the minimax game under a covariate shift assumption.
>
> **Regarding the “why this is the right formulation?” and the ablation study:**
>
> * We do have ablation studies in terms of different ways to do the density ratio estimation and uncertainty quantification. Within our formulation, we set the ratio to 1 for all the data points ($R = 1$) and we also set the label regularization to 0 ($r=0$), see figure 3 (a) and (b) in the new version. For comparison with other uncertainty measures, we compare with AVH scores for self-training (See Table 1). AVH score is an alternative “instance hardness” metric that can be used for self-training. Our proposed DRST outperforms all the baselines in the ablation study. We added detailed descriptions about the ablation to Section 3.1.
>
> * It is a very good suggestion to use a standard UDA method to replace DRL. We did this experiment using DeepCoral[Sun and Kate, 2016] + CBST. DeepCoral is a standard UDA method based on feature alignment. We obtain 82.39% accuracy on VisDA, still almost 3% lower than DRST. We added this new baseline to the paper (See Table 1).
>
>
> **Regarding “How does a different choice for the flexibility of the adversary affect UDA results (Eq.3)”:**
>
>  It is a good question of how the constraint set for the adversary affects the results. In the original DRL, the constraint set can be constructed using linear, quadratic, higher-order features, or even kernel functions with different slack terms. In principle, the more flexibility the adversary has, the more flexible the resulting model is. For example, if the adversary is constrained tightly with high-order moments or large dimensions of features, the resulting method is more likely to overfit. Also, the larger the slack term in the constraint set is, the more regularized the resulting model is. In this paper’s case, we learn the features directly from data using neural networks. This avoids the feature choice issue. The slack term choice is also translated to the weight decaying hyperparameter choice. However, in our experiment, we found different weight decay is not affecting the result much.
>
> **Regarding the $p$ proportion in self-training:**
>
>  In DRST, we set $p = 0.5$ and fix it throughout the self-training. We added this to the paper. The effects of $p$ proportion have been investigated thoroughly in the CBST paper [Zou et al. 2018]. It was shown there that p proportion does not make a big difference in the results. In our case, we have tried the recommended $p$ proportion in CBST and found the results to be similar to our current strategy. We then use $p = 0.5$ for simplicity.
>
> References:
>
> [Sun and Kate, 2016] Sun, Baochen, and Kate Saenko. "Deep coral: Correlation alignment for deep domain adaptation." European conference on computer vision. Springer, Cham, 2016.
>
> [Zou et al. 2018] Zou, Yang, et al. "Unsupervised domain adaptation for semantic segmentation via class-balanced self-training." Proceedings of the European conference on computer vision (ECCV). 2018.

---

### Official Review · AnonReviewer3 · 2020-10-27
**submission 2815 review**

**Rating:** 5
**Confidence:** 3

**Review:**

summary:
This paper introduces distributional robust learning (DRL) for the unsupervised domain adaptation task. The proposed DRL based approach is claimed to better measure model prediction during self-training with an auxiliary discriminator. This paper also recognizes an optimization issue of conventional DRL and proposes an end-to-end solution via density ratio estimation.

However, I have the following concerns,

1.The generalization error (both theoretically and empirically) of the gradient approximation is unclear. It is necessary to analyze how effective and under what conditions the proposed approximation can work for the expected target loss optimization.

2.It needs elaboration why the density ratios can be directly replaced as discriminator predictions, which seems not straight-forward and is the main difference to the conventional DRL.

3.There are some related work not covered in the paper. It is quite surprising that Ben-Tal et al. (2013) is not discussed, also a few later application work (Shafieezadeh-Abadeh et al. 2015; Sinha, Namkoong and Duchi 2018; Hu et al. 2018). More importantly, some existing study already touched high-dimension data and self-training (19NIPS). A thorough discussion of how this work makes unique contributions is required.

4.The baseline results on Office31 and Office-Home datasets are missing. Besides, since it claims to better measure prediction uncertainty, a direct comparison of prediction confidence with the baselines could better justify the superiority of the proposed approach.


Ben-Tal et al. "Robust solutions of optimization problems affected by uncertain probabilities," Management Science, 2013
Shafieezadeh-Abadeh et al., “Distributionally robust logistic regression,” in NeurIPS, 2015
Sinha, Namkoong and Duchi, “Certifiable distributional robustness with principled adversarial training,” in ICLR, 2018.
Hu et al., “Does distributionally robust supervised learning give
robust classifiers?” in ICML, 2018
Najaf et al., "Robustness to Adversarial Perturbations in Learning from Incomplete Data," NeurIPS, 2019

---

> ### Author Response · Authors · 2020-11-23
> **Author response to AnonReviewer3**
>
> Thanks for your detailed comments! Regarding the gradient approximation and the missing office results, please kindly refer to our general response. We now address your additional concerns:
>
> **Regarding using a discriminator as a density ratio estimation:** We added references for discriminative learning for two distributions in the paper. Specifically, as mentioned briefly in the paper, according to the Bayes’ rule, the density ratio estimation on $\frac{P_s(x)}{P_t(x)}$ can be converted to: $\frac{P(s|x)P(t)}{P(t|x)P(s)}$, where $\frac{P(s|x)}{P(t|x)}$ can be viewed as the output probability of a discriminator. We also added the full derivation and make the connection more clear in the paper. For more details, we refer to this paper: Bickel, Steffen, Michael Brückner, and Tobias Scheffer. "Discriminative learning for differing training and test distributions." Proceedings of the 24th international conference on Machine learning. 2007.
>
> **Regarding the missing citations:** Thanks a lot for the suggestions on the additional citations. We added them in the related work in the distributionally robust learning subsection. However, it is worth noting that there is a significant difference between our method and the methods in the suggested references. Our adversary is perturbing the conditional labeling function, not the covariate variables. This leads to a very different algorithm from the suggested works. In particular, different from the Neurips19 paper, we are not focusing on adversarial perturbations. Instead, we show that by using DRL, we avoid overconfidence in prediction, which benefits self-training UDA significantly. The derivation of our parametric form and the integration of the density ratio estimation in an end-to-end fashion are novel and unique. We added explanations to the paper on how these two distributionally robust methods differ and connect with each other.

---

### Official Review · AnonReviewer2 · 2020-11-01
**Promising approach to improve self-training for unsupervised domain adaptation**

**Rating:** 7
**Confidence:** 4

**Review:**

**Summary**
The paper proposes to use the distributionally robust learning (DRL) for unsupervised domain adaptation. First, the authors demonstrate how differentiable density ratio estimation can be done for source and target domains in an end-to-end manner. Following this, the authors demonstrate how confidence estimation (reliant on DRL) can be utilized for self-training based approaches for unsupervised domain adaptation. In terms of results, the authors show that the proposed approach — Distributionally Robust Self-Training (DRSL) — can provide competitive performance on t he VisDA 2017 benchmark. Furthermore, DRSL is able to achieve more calibrated probabilities (useful for more calibrated uncertainty measures) with competitive predictive performance. Finally, the authors provide some intuition as to why DRSL results in increasingly aligned conditional source and target distributions due to increased focus on learning shape features.


**Strengths**
- With the exception of a few points raised in the weaknesses section, the paper is generally well-written and easy to follow.
- When initialized with ASG, the proposed approach shows roughly ~1% improvement compared to baselines and prior approaches on the VisDA benchmark. This improvement is coupled with increased sample-efficiency (significant within 95% standard error), more calibrated probabilities and monotonic reductions in the measure distribution gap. This is perhaps one of the strongest positive points in support of the proposed approach.
- The presented analysis about the kind of features DRST increasingly focuses on (although not fleshed out more extensively) is interesting and can likely provide more insights into how to improve on DRST in future work.

**Weakesses**
- One of the minor weaknesses with the current draft is readability. I found it slightly hard to digest the proposed approach in the form it is presented in the current draft. For instance, it was unclear as to how gradient expressions for the classification network from the expected target loss for DRL from equation 8 were aligned with the proposed reduction in equation (6) — specifically, what data are the gradients estimated on? In context of DRST, is this on labeled source + pseudo labeled target per-epoch from an earlier DRST epoch? While sections 2.1 and 2.2 in themselves are more or less straightforward to follow, putting them together in context of DRST seems slightly hard for a reader. Improving this in terms of clarity would definitely benefit the paper.
- While the experiments attempting to understand the kind of features DRST focuses compared to baselines and other approaches using Grad-CAM are useful and definitely provide some insight, the paper would definitely benefit if this claim were backed by quantitative metrics of some kind. Namely, (1) to understand if DRST indeed focuses on shape features, what is the mean IoU of region highlighted by grad-cam (w.r.t. the predicted class) with (say) the segmentation mask of concerned object in the image and (2) since Grad-CAM is being relied on as a tool to obtain saliency maps, observed saliency maps should also be provided with measures of fidelity — how reliant are Grad-CAM maps in terms of providing feature importance visualizations in context of the models and dataset involved (see [A] for adopted benchmarks and metrics, namely the Remove and Re-train (ROAR) metric). Since focus on shape features if proposed as a cause for declining source and target conditional distributions, including measures from the aforementioned points would strengthen this claim.

[A] - A Benchmark for Interpretability Methods in Deep Neural Networks

---

> ### Author Response · Authors · 2020-11-23
> **Author response to  AnonReviewer2**
>
> Thanks for the detailed comments and suggestions! We appreciate your nice summary of our contributions.
>
> **In terms of further justifying that our method captures better shape features:**
>
> (1) Thanks for the suggestion on further validating our points on capturing the shape features. We agree that providing more quantitative evaluation on the shape features would strengthen our claim. Showing the mean IoU is a very good idea. However, to our best knowledge, there is no ground truth segmentation masks for VisDA since it is cropped from the COCO dataset. In an attempt to further provide a quantitative metric, we resort to a similar dataset (LID: https://lidchallenge.github.io/challenge.html) -- an annotated subset of the imagenet data that was used for weakly-supervised semantic segmentation. We believe this data is similar enough with VisDA for the same categories. We select categories covered by both data and infer directly on the LID validation set using our trained model from VisDA. In particular, we compare Grad-CAM’s feature map in the ground truth labels with the ground truth segmentation masks and calculate the IoU. We list the results as follows:
>
> | Class:  | Airplane | Bicycle | Bus   | Car    | Horse| Motorcycle| Person |Train  | Average|
> | ---         |---             |---          | ---      | ---      | ---       |---                 |---           | ---      | ---           |
> | CBST:  | 0.354       | 0.296    |0.525 | 0.358| 0.342  | 0.469          | 0.307    | 0.237 | 0.361      |
> |DRST:| 0.348 |**0.313**| 0.496| 0.348| **0.365**| 0.462| **0.312**| **0.262** |**0.363**|
>
> We can see that DRST outperforms CBST in half of the categories and has an advantage on average.
>
> (2) We also appreciate the suggestion on evaluating the reliability of Grad-CAM as a measurement of feature importance. We understand the concern that Grad-CAM may not be the best method for showing the attention of the model. However, to the best of our knowledge, Grad-CAM is widely-used and well-accepted for the single-object-centric image classification tasks and the domain adaptation community [Chen et al. 2020, Wen et al. 2020, Li et al. 2020]. Our results on Grad-CAM gives us important hints on why our method performs better. We leave the investigation of different feature importance metrics for interpretability under domain shift as an important future work.
>
>
> References:
>
> [Chen et al. 2020] "Automated synthetic-to-real generalization." arXiv preprint arXiv:2007.06965 (2020).
>
> [Wen et al. 2020] "Interventional Domain Adaptation." arXiv preprint arXiv:2011.03737 (2020).
>
> [Li et al. 2020] "Learning Invariant Representations and Risks for Semi-supervised Domain Adaptation." arXiv preprint arXiv:2010.04647 (2020)

---

### Author Response · Authors · 2020-11-23
**General Response to All the Reviewers**

We thank all the reviewers for their detailed comments and suggestions! We appreciate that our contributions are recognized.

**TLDR**: Our paper is the first to design practical approaches for uncertainty quantification (UQ) in deep neural networks using a principled foundation of distributional robustness. We achieved state-of-art results in unsupervised domain adaptation by combining our UQ with self-training.

We now address some common concerns among reviewers. We will reply to individual comments for specific questions. We also utilize the extra page and update the paper according to comments and suggestions.

* **Regarding the formulation of DRL (Figure 1(a)(b) and Figure 2) (R2, R3, R4)**:

    Our final form of distributionally robust learning (DRL) is an instantiation of the general distributionally robust formulation in Figure 1(a) using neural networks. In particular, the distributionally robust formulation under covariate shift generates a parametric form that consists of a density ratio. It can be implemented by a classification network and a discriminative density ratio estimation network, as in Figure 1(b). The final instantiation then utilizes the formulation in Figure 1(b) and an additional class-level regularization. The class-level regularization is derived directly from a regularized objective in the distributionally robust formulation. Figure 2 summarizes the incorporation of DRL in the self-training process.

* **Regarding gradient (of the classification network) computation (related questions asked by all the reviewers)**:

    * In general, the gradient for the classification network is computed on labeled source data. The reason is, under our setting, the gradients of the expected target loss is only related to source distribution feature expectation, due to the density ratio in the parametric form (See equation 6). So we are just using empirical statistics in the source to approximate expected statistics in the same distribution. (**R4**)

    * The 2-norm of the approximated gradient converges to the true one theoretically at the rate of $O(1/m)$, where “m” is the number of source data. We revised the paper accordingly. This is not a novel result but is from the previous DRL work that used low-dimension features (See Eq. 3.37 in  [Liu, 2018]). (**R3**)

    * In terms of the specific equations for gradients, it is a very good question how (6) and (8) are aligned. (**R2**) We added a brief explanation after (8). In short, the gradient of $\theta$ in (6) is equivalent to the gradient for $w$ in (8) (the second equation), when $\Phi$ is set to be $y\phi(x, \alpha)$. More specifically, $\hat{P}(y|x)$ in (6) is the ${\bf \hat{P}}$ in (8), which is our prediction. The $\tilde{P}(y|x)$ in (6) is $\bf y$ in (8), which is the empirical label data. We only need the gradient for $\theta$ in the original DRL formulation for learning, while we need gradients for $\phi$ and $b$ as well for deep learning. So the gradient for $\phi$ and offset $b$ is derived from the expected target loss similarly.  (**R2**)

    * In the context of self-training, we have a “new source” for each epoch, which is precisely what R2 has mentioned -- the labeled source + pseudo labeled target (See Figure 2). We then use this “new source” to calculate the gradient.  We edited Section 2.3 to put our method in the context of self-training in Section 2.3. We also put Figure 2 closer to Section 2.3. (**R2**)


* **Missing shape features analysis (R4)**:

    We provided the shape feature visualization in Figure 4. We moved it closer to the experiment section such that it is closer to the context. More visualization on the shape features captured by DRST is also provided in Figure 10 in the appendix.

* **Missing office data results (R3)**:

    We had experiments from the Office dataset before the deadline. However, due to the page limit, we only had space for a brief discussion about the results from these two datasets in Section 3.2. In Section 3.2, we use office-31 and office-Home data to demonstrate the effectiveness of DRL in improving learning accuracy and the level of calibration (See Figure 6). Now, given the one extra page, we add more results on office data to the paper and analyzed the results more thoroughly in Section 3.2. We also edited the general description of our experiments so that the purpose of each set of experiments is more clear.


References: [Liu, 2018] Robust Prediction Methods for Covariate Shift and Active Learning. Thesis.

---

### Decision · Program_Chairs · 2021-01-07
**Final Decision**

**Decision:**

Reject

**Comment:**

The paper proposes a rather complex algorithm for unsupervised doamin adaptation.
While the paper provides detailed explanation, some motivation and some experimental resulst,
it does not provide any theoretical guarantees for its performance. More concerning, since domain adaptation
can only succeed when there is a close relationship between the source and target tasks, and only with algorithms
that take that relationship into account, any scientific proposal for domain adaptation should include a clear
discussion of the assumptions driving the proposed algorithms and of the circumstances under which the proposed approach
may or may not work. This is missing in the current submission.

More specifically, a similar ocncern was voiced by Reviewer 3
Namely ".The generalization error (both theoretically and empirically) of the gradient approximation is unclear. It is necessary to analyze how effective and under what conditions the proposed approximation can work for the expected target loss optimization." Thsi point was not addressed in teh authors' rebuttal.

Anotehr key concerning point that was also brought up by reviewer 3 read:
"It needs elaboration why the density ratios can be directly replaced as discriminator predictions, which seems not straight-forward and is the main difference to the conventional DRL." In response the authors cite the paper by Bickel et al 2007 but it falls short of addressing the well know fact that density ratio cannot be reliably estimated from samples of bounded size. The authors should have explained specific assumptions that can make this step of their algorithm og through.